# A framework for reconstructing SARS-CoV-2 transmission dynamics using excess mortality data

Mahan Ghafari [1,5 ✉], Oliver J. Watson [2,5], Ariel Karlinsky [3], Luca Ferretti[4] & Aris Katzourakis[1 ✉]

The transmission dynamics and burden of SARS-CoV-2 in many regions of the world is still largely unknown due to the scarcity of epidemiological analyses and lack of testing to assess the prevalence of disease. In this work, we develop a quantitative framework based on excess mortality data to reconstruct SARS-CoV-2 transmission dynamics and assess the level of underreporting in infections and deaths. Using weekly all-cause mortality data from Iran, we are able to show a strong agreement between our attack rate estimates and seroprevalence measurements in each province and find significant heterogeneity in the level of exposure across the country with 11 provinces reaching near 100% attack rates. Despite having a young population, our analysis reveals that incorporating limited access to medical services in our model, coupled with undercounting of COVID-19-related deaths, leads to estimates of infection fatality rate in most provinces of Iran that are comparable to high-income countries.

---

[1] Department of Zoology, University of Oxford, Oxford, UK. [2] Department of Infectious Disease Epidemiology, Imperial College London, London, UK.
[3] Department of Economics, Hebrew University of Jerusalem, Jerusalem, Israel. [4] Big Data Institute, Li Ka Shing Centre for Health Information and Discovery, Nuffield Department of Medicine, University of Oxford, Oxford, UK. [5]These authors contributed equally: Mahan Ghafari, Oliver J. Watson. ✉email: mahan.ghafari@zoo.ox.ac.uk; aris.katzourakis@zoo.ox.ac.uk

Assessing the true burden of COVID-19 requires a comprehensive surveillance system to monitor outbreaks and a large capacity for diagnostic testing. However, this is particularly challenging for many low- and middle-income countries due to limited diagnostic capacities and access to healthcare[1–6]. An alternative approach to measuring the burden of COVID-19 is to monitor changes in all-cause mortality trends with respect to baseline levels based on historic trends (i.e., measuring excess mortality). While excess mortality during the current pandemic may not exactly match the true number of COVID-19-related deaths due to various reasons such as disruptions in treatment for other fatal diseases and socio-economic disparities in health care in different regions of a country, several works have shown that a significant portion of excess deaths in many countries are directly attributable to COVID-19[1,7–11]. This enables us to use excess mortality as a proxy for estimating the number of infections and fatalities from COVID-19 and correct for under-reporting of cases and deaths that may arise due to limited testing from suspected cases and uncertainty in the number of fatalities attributable to COVID-19[12,13].

Iran was among the first countries outside mainland China to report a large outbreak of SARS-CoV-2 in February and March 2020 and also acted as a major source for its spread in several countries in the Middle East and elsewhere[11,14–18]. It was also one of the first countries to experience a second wave of infection after the relaxation of non-pharmaceutical interventions (NPIs) in July 2020[14,19]. By the end of year 2021, the country experienced three additional waves, two of which were driven by the Alpha and Delta variants of SARS-CoV-2, in May and September 2021. Despite a recent increase in the number of daily administered vaccine doses, Iran had a very slow start on its immunisation programme with only ~3% of the population being fully vaccinated by the end of July 2021, during the surge in cases with the Delta variant[20]. Because the Ministry of Health and Medical Education (MoHME) of Iran stopped releasing province-level data on the number of confirmed COVID-19 cases and deaths from 2020-03-22, the transmission dynamics across the country have remained largely unknown. One way to estimate the number of infections and account for under-reporting of cases is by converting deaths attributable to COVID-19 to infections using a population-weighted infection fatality ratio (IFR) estimate (i.e., infections = deaths/IFR), taking into account the demographics of the population of interest[21]. However, such estimation of attack rates can be inaccurate or misleading if it does not take into account the impact of limited access to health care in municipalities with low socioeconomic status, uneven adoption of pharmaceutical and non-pharmaceutical intervention in different areas, and the increased chance of re-infection over time. In particular, it is clear from the SARS-CoV-2 epidemics arounds the world during 2020 and 2021 that the demand for intensive care beds and mechanical ventilators can place a severe strain on health systems. This has likely more profound consequences for low-income settings where the availability and quality of healthcare and related resources is typically limited[22]. Neglecting to account for these factors can result in an overestimate of SARS-CoV-2 attack rates and underestimation its potential for continued transmission, with immediate implications for assessments of the potential burden of SARS-CoV-2 variants of concern such as Omicron in a population.

In this work, we use the newly updated province-level age-stratified weekly all-cause mortality data from the National Organization for Civil Registration (NOCR) of Iran and develop a mathematical framework to fully reconstruct the Iranian epidemic with the aim to estimate the attack rate, number of daily hospital admissions, deaths, and re-infection rates of SARS-CoV-2 across the country.

## Results

**Estimating province-level excess mortality in Iran.** We obtain age-stratified weekly all-cause mortality data from NOCR and apply a statistical model to calculate the weekly excess mortality for all 31 provinces and across 17 age-groups during the Iranian epidemic (see Methods section). Figure 1a shows 5 distinct peaks in excess mortality with a temporal trend that is strongly associated with the nationwide reported COVID-19 deaths over time. Several provinces show significant levels of excess mortality from the first week of February 2020 suggesting that the Iranian epidemic likely started at least a month before that (Fig. 1b). This is in agreement with previous phylogenetic and epidemiological studies suggesting the epidemic started in late December 2019 to early January 2020[14,23]. Furthermore, during the first two waves of the Iranian epidemic, the excess mortality was roughly 2.5 times higher than reported COVID-19 fatalities. However, this dropped to 2.1 at the later stages suggesting that the testing capacity of the country to record infections and deaths gradually improved over time, but not to the extent to fully capture the majority of COVID-19-related deaths. This is also in agreement with statements from the Iranian health officials suggesting that Iran's COVID-19-related deaths could be twice the official numbers and also previous studies showing elevated levels of under-reporting of COVID-19 fatalities in Iran, particularly in the early stages of the pandemic[14,15]. We also find that the cumulative nationwide excess deaths by 2021-10-22 is 259,641 (95% CI: 230,493–288,790) which is nearly twice the 125,000 reported COVID-19 deaths at the time.

Figure 1b shows the intensity of excess mortality as quantified by the number of excess deaths per 100,000 persons in each province over time. During the first wave in spring 2020, only a few provinces showed significant levels of excess mortality with Gilan, Qom, Mazandaran, and Golestan among the hardest-hit provinces. This is likely because these provinces usually attract a large number of tourists and pilgrims around the New Year's holiday in February/March which led to large superspreading events and uncontrolled outbreaks due to limited NPIs before the national lockdown on 2020-03-05[14]. As the epidemic progressed, the timing of the peak excess mortality between different provinces were more simultaneous (Fig. 1b; Supplementary Fig. 1). Most notably, during the third wave in October/November 2020 and the fifth wave in August/September 2021 with the Delta variant of SARS-CoV-2, almost all provinces experienced significant number of fatalities around the same time.

**Estimating excess mortality per age-group.** We also carry out a similar analysis to estimate the temporal pattern of COVID-19 fatalities for different age-groups (Fig. 1c). We find that the majority of excess deaths are concentrated in the older age categories (>55) and that the third and fifth epidemic waves had the largest impact on younger age groups. Figure 1d shows the per capita ratio of excess mortality per age group with respect to the 40–49 age group. The reason for choosing the 40–49 age group as a baseline is because, unlike younger age groups, excess mortality in this age group is subject to relatively low fluctuations over time due to experiencing more deaths. It also serves as a baseline for measuring the impact of vaccination on lowering excess deaths in older age groups who have received vaccination at an earlier date (see Supplementary Fig. 2). We see that while there is a gradual increase in the ratio of deaths in the older age-groups during the first three waves, there is a two-fold drop in the 60–74 and 75+ age-groups before the start of the fifth wave. This is likely due to a higher vaccination coverage in these age-groups. While the exact number of vaccinated individuals per age-group in Iran is not known, given that the immunisation programme for

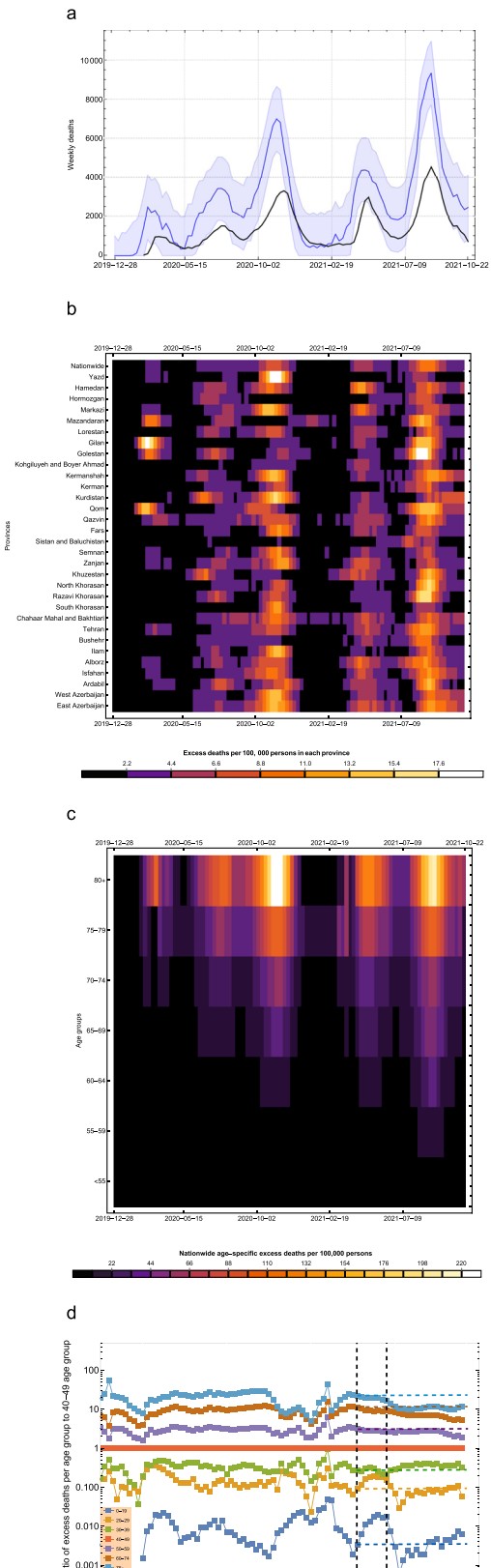

**Fig. 1 Estimating excess mortality per province and age-group during the pandemic in Iran. a** One-month average of weekly excess mortality (blue) and reported COVID-19 deaths (black) in Iran from 2019-12-28 to 2021-10-22. Central line and shaded area show the median and 95% forecast intervals in weekly excess mortality, respectively. **b** One-month average of weekly excess deaths per 100,000 persons per province over time. **c** Nationwide excess deaths per 100,000 persons per age group over time. **d** Ratio of weekly excess deaths per age-group to the 40–49 age-group. Vertical dashed lines show the start of the immunisation programme for individuals aged 75 and above on 2021-05-18 and 60 and above on 2021-07-14. The horizontal dashed lines show the ratio of excess deaths at the start of the immunisation programme for individuals aged 75 and above.

drop in excess mortality in the older age-groups before and during the fifth wave is due to their high vaccination coverage.

**Model validation.** Assuming that lack of testing is the primary reason behind the discrepancy between excess mortality and official COVID-19 deaths[1,15,24,25], we attribute the mean excess mortality during the pandemic period to COVID-19. This enables us to reconstruct the SARS-CoV-2 transmission dynamics across the country and infer the population fatality rate (PFR), IFR, and attack rate per province (see Methods section). We first estimate the national and provincial attack rates of SARS-CoV-2 under two different sets of assumptions of the IFR by age relationship and select the better-fitting model by comparing the modelled seroprevalence estimates to data (Supplementary Fig. 3; Supplementary Data 1–6). By tracking the number of people who require a general hospital or ICU bed over time, we were able to account for the impact of bed shortages on increasing IFR for COVID-19 patients who do not receive a bed in each province (Methods section). We find that the model with a higher IFR per age-group (and an overall lower estimated attack rate) that explicitly accounts for an increase in IFR when hospital capacity is reached provides a better fit to seroprevalence data. With a median value of approximately 0.50% (0.32–0.64%), our inferred province-level IFR is generally higher than a population-weighted IFR which only takes into account the demographics of a given province (Table 1; Methods section) and is roughly twice as high as the IFR estimates derived from serological studies in predominantly high-income countries[21]. This discrepancy is expected because disparities in healthcare access can be offset by the younger population age structure of many developing countries resulting in population IFRs that are similar to high-income countries[7,26,27]. Figure 2a, b show a nearly 75% attack rate of SARS-CoV-2 across the country with nearly 15% of all infections due to reinfections by 2021-10-22. We also find that the age-distribution of the modelled COVID-19 deaths are well-aligned with the pattern of excess mortality per age-group and both follow a clear log-linear relationship with respect to age, a signature characteristic of COVID-19 fatalities[28], suggesting that the model successfully reconstructs age-dependent contact patterns for disease spread in Iran (Fig. 2c). There is also a strong association between the modelled and observed daily hospital admissions over time and across provinces, further indicating that the majority of excess deaths during the pandemic period are directly attributable to COVID-19 (Supplementary Fig. 4). There is, however, a notably lower number of modelled daily hospital admissions compared to observed hospitalisations in some provinces (e.g., South Khorasan, Lorestan, and Yazd) during the last wave in comparison to earlier waves. To explore these patterns, we use a mixed-effects model to examine the relationship between the number of hospitalisations inferred from our model fits and the observed COVID-19 hospital admissions over time and across

individuals aged >75 started in May 2021 (first phase priority groups for vaccination in Iran are frontline healthcare workers as well as those aged 80 and above; see[20] for more information), and that Iran fully vaccinated roughly 5 times the population size of individuals aged 75 and above by 2021-09-07, it is likely that the

**Table 1 The cumulative excess mortality, PFR, estimated IFR (estimated from the transmission model), population-weighted IFR (based on the age-distribution of each province), and percentage of the population ever exposed to SARS-CoV-2 in each province as of 2021-10-22.**

| Province | Excess deaths | % PFR | % Estimated IFR | % Population-weighted IFR | % Exposed |
|---|---|---|---|---|---|
| East Azerbaijan | 17,035 (8,887–28,237) | 0.438 (0.228–0.726) | 0.50 (0.41–0.63) | 0.41 (0.29–0.53) | 84.92 (69.06–103.05) |
| West Azerbaijan | 11,609 (6289–19,204) | 0.346 (0.187–0.573) | 0.47 (0.36–0.57) | 0.33 (0.23–0.43) | 75.66 (60.25–98.53) |
| Ardabil | 4,538 (1974–8053) | 0.361 (0.157–0.641) | 0.38 (0.31–0.46) | 0.37 (0.26–0.47) | 92.55 (76.19–112.14) |
| Isfahan | 18,281 (9612–30,412) | 0.355 (0.187–0.591) | 0.52 (0.41–0.65) | 0.41 (0.29–0.53) | 65.90 (51.80–81.46) |
| Alborz | 11,068 (5477–18,219) | 0.388 (0.192–0.639) | 0.49 (0.38–0.61) | 0.35 (0.24–0.46) | 64.09 (51.07–81.97) |
| Ilam | 1719 (611–3743) | 0.294 (0.104–0.640) | 0.47 (0.38–0.56) | 0.32 (0.22–0.42) | 61.35 (51.01–73.74) |
| Bushehr | 3023 (1241–5,649) | 0.244 (0.100–0.45) | 0.43 (0.31–0.55) | 0.26 (0.17–0.35) | 61.29 (44.63–82.10) |
| Tehran | 49,320 (27,155–78,900) | 0.365 (0.201–0.585) | 0.47 (0.39–0.58) | 0.40 (0.28–0.52) | 77.06 (65.08–93.99) |
| Chahaar Mahal & Bakhtiari* | 3624 (1089–7452) | 0.376 (0.113–0.774) | 0.49 (0.38–0.61) | 0.33 (0.23–0.43) | 53.05 (43.49–73.44) |
| South Khorasan | 1736 (473–4466) | 0.210 (0.059–0.558) | 0.50 (0.43–0.59) | 0.36 (0.26–0.46) | 41.27 (34.65–48.50) |
| Razavi Khorasan | 20,847 (9945–36,946) | 0.300 (0.147–0.549) | 0.36 (0.29–0.43) | 0.33 (0.23–0.42) | 90.19 (71.76–113.32) |
| North Khorasan | 2478 (985–5250) | 0.283 (0.112–0.600) | 0.36 (0.27–0.44) | 0.32 (0.22–0.42) | 77.73 (61.94–100.78) |
| Khuzestan | 15,366 (8235–25,189) | 0.317 (0.170–0.520) | 0.32 (0.26–0.40) | 0.28 (0.19–0.37) | 98.48 (79.82–120.59) |
| Zanjan | 3938 (1442–7797) | 0.367 (0.134–0.720) | 0.50 (0.42–0.61) | 0.38 (0.27–0.49) | 73.11 (60.85–87.84) |
| Semnan | 2057 (742–4226) | 0.275 (0.099–0.566) | 0.49 (0.43–0.58) | 0.37 (0.26–0.48) | 57.26 (49.85–65.92) |
| Sistan & Baluchistan | 4401 (1104–12,829) | 0.140 (0.036–0.419) | 0.37 (0.25–0.49) | 0.18 (0.12–0.24) | 46.60 (35.38–67.60) |
| Fars | 13,825 (6361–25,441) | 0.282 (0.130–0.520) | 0.50 (0.40–0.62) | 0.36 (0.25–0.47) | 56.63 (42.64–71.82) |
| Qazvin | 5255 (1793–9699) | 0.404 (0.138–0.747) | 0.47 (0.38–0.59) | 0.35 (0.24–0.45) | 85.97 (69.52–106.44) |
| Qom | 5519 (3193–8925) | 0.399 (0.231–0.646) | 0.49 (0.33–0.62) | 0.30 (0.20–0.39) | 84.74 (66.12–116.36) |
| Kurdistan | 6421 (3015–11,243) | 0.394 (0.185–0.69) | 0.47 (0.37–0.58) | 0.34 (0.24–0.45) | 80.05 (66.82–103.86) |
| Kerman | 7340 (2761–14,621) | 0.224 (0.084–0.446) | 0.46 (0.36–0.57) | 0.30 (0.21–0.40) | 50.58 (40.64–63.22) |
| Kermanshah | 6844 (3662–11,734) | 0.356 (0.190–0.610) | 0.51 (0.39–0.63) | 0.38 (0.27–0.50) | 66.67 (54.07–86.87) |
| Kohgiluyeh & Boyer Ahmad | 1417 (301–3569) | 0.191 (0.040–0.481) | 0.32 (0.26–0.39) | 0.28 (0.19–0.37) | 59.41 (48.25–74.93) |
| Golestan | 6920 (3138–12,352) | 0.357 (0.162–0.630) | 0.35 (0.27–0.43) | 0.30 (0.20–0.39) | 105.43 (87.31–125.82) |
| Gilan | 8788 (3888–17,078) | 0.362 (0.160–0.704) | 0.64 (0.51–0.75) | 0.50 (0.36–0.65) | 52.62 (45.21–67.60) |
| Lorestan | 4942 (1895–9850) | 0.280 (0.100–0.566) | 0.48 (0.40–0.59) | 0.35 (0.24–0.45) | 56.51 (47.10–68.36) |
| Mazandaran | 10,500 (4372–19,563) | 0.324 (0.130–0.603) | 0.55 (0.46–0.64) | 0.44 (0.31–0.57) | 58.36 (48.01–69.78) |
| Markazi | 5200 (2273–9601) | 0.367 (0.160–0.678) | 0.55 (0.46–0.67) | 0.43 (0.31–0.55) | 64.71 (53.01–75.59) |
| Hormozgan | 3753 (1369–7864) | 0.194 (0.070–0.407) | 0.42 (0.33–0.54) | 0.24 (0.16–0.32) | 48.52 (38.76–64.10) |
| Hamedan | 5952 (2868–10,475) | 0.300 (0.168–0.615) | 0.53 (0.44–0.65) | 0.41 (0.29–0.53) | 63.97 (53.45–76.72) |
| Yazd | 3287 (1572–6475) | 0.270 (0.129–0.533) | 0.35 (0.30–0.40) | 0.33 (0.23–0.43) | 80.50 (71.43–93.68) |

*Year 1394 Solar Hijri is excluded from the calculation of background mortality for this province due to a record-high excess mortality in that year[15].

epidemic waves while controlling for province-specific variation (see Methods section). Using this framework, we show that there was no significant decrease in the ratio of model inferred hospitalisations to observed hospitalisations during the last wave compared to previous waves, and that the majority of variation in hospitalisation patterns observed is explained by province-level random effects, most likely due to specific changes in treatment seeking or the age demographics being most affected in later waves (Supplementary Fig. 5). However, a significant decrease in the ratio of model inferred hospitalisations to observed hospitalisations was observed since the start of the pandemic, which suggests that case fatality rates from hospitalised infections have decreased during this period, likely due to a combination of improved case management, increased number of infections resulting from reinfections, and higher vaccination coverage.

**Estimating cumulative attack rate of SARS-CoV-2.** Using our transmission model, we also calculate the cumulative attack rate by 2021-10-22 for each province (Fig. 2d). Our results show a high degree of heterogeneity in the estimated attack rates across the country with 11 provinces reaching close to and higher than 100% cumulative attack rates after the fifth wave (Supplementary Table 1). While in most regions, the majority of infections took place during the third and fifth waves, the Delta variant of SARS-CoV-2 was responsible for the highest re-infection rates during

the fifth wave (Supplementary Fig. 6). Qazvin, Qom, and East Azerbaijan had the highest per capita mortality rate with cumulative PFRs of nearly 0.4%, while Golestan, Khuzestan, and Qom had the highest re-infection rate with nearly 20% of infections being due to re-infection (Table 1; Supplementary Fig. 6). Figure S1 and Supplementary Fig. 4 also show that while some provinces such as Ardabil, Alborz, Semnan, Golestan, Hamedan, and Mazandaran experienced five distinct epidemic peaks, others like Bushehr, Qazvin, Gilan, Kohgiluyeh and Boyer Ahmad, and Hormozgan experienced sustained transmission of SARS-CoV-2 for an extended period of time throughout the pandemic. In late September 2021, official reports from MoHME indicate that COVID-19 deaths in Sistan and Baluchistan, Ardabil, Ilam, Bushehr, Chahaar Mahal and Bakhtiari, South Khorasan, North Khorasan, Zanjan, Kohgiluye and Boyer Ahmad, Lorestan, Hormozgan, and Hamedan have all dropped to below 5 individuals per day in recent weeks, an almost record-low since the first wave in March 2020[29–31]. This is congruent with our estimates of high attack rates in these provinces which may have resulted in a (temporary) herd-immunity to be reached after the fifth wave.

**Discussion**

Since the start of the pandemic, the global heterogeneity in COVID-19 reporting systems globally has resulted in substantial debate about the true extent to which COVID-19 has spread in

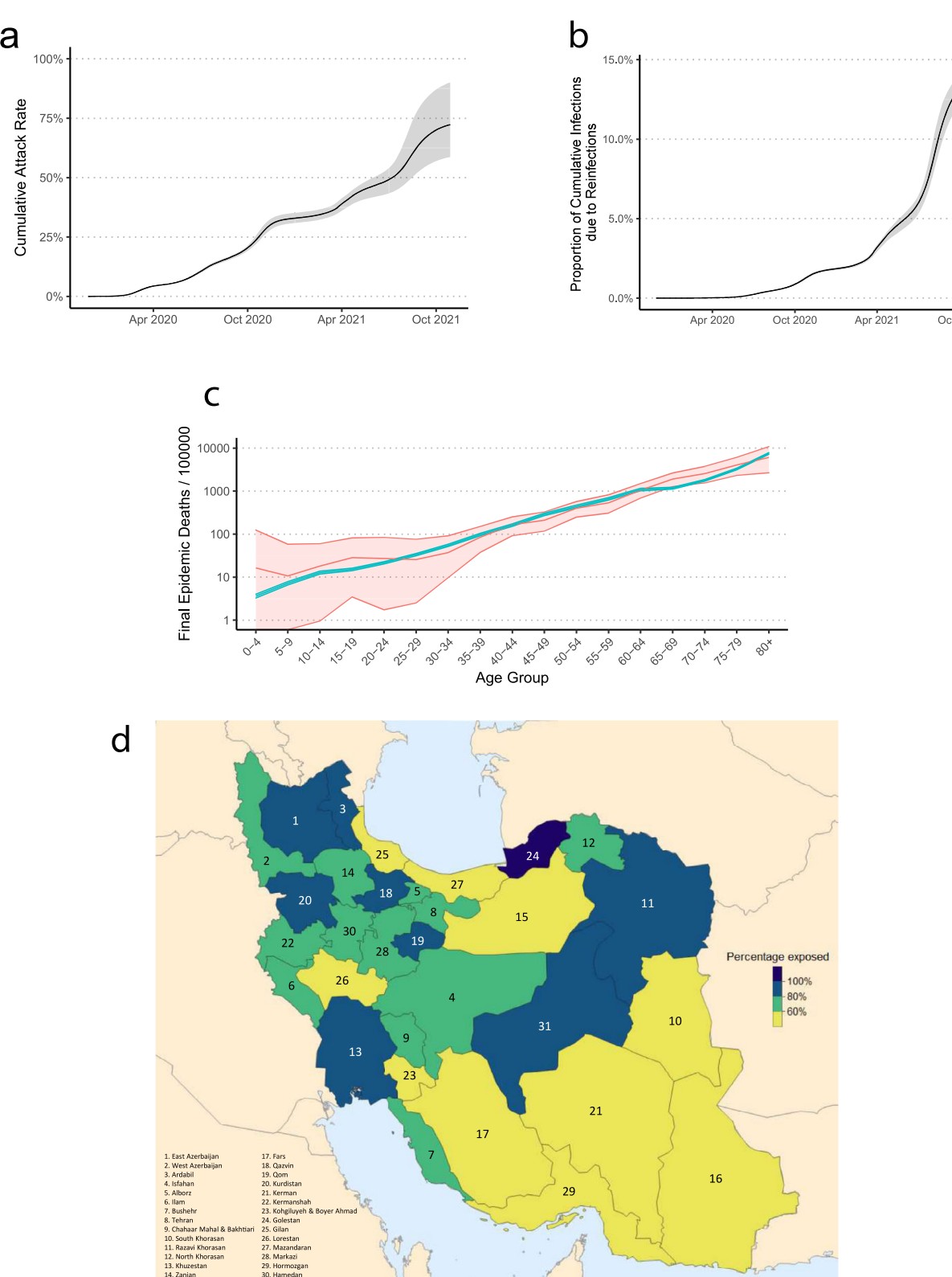

**Fig. 2 Estimating cumulative attack rates and reinfection rates of SARS-CoV-2 in Iran. a** Modelled cumulative nationwide attack rate of SARS-CoV-2 and **b** percentage of individuals reinfected over time. In both a and b, shaded area in black shows the estimated quantities from the three sets of model assumptions used for estimating attack rates. The central line shows the central scenario and bands show the range reflected by the optimistic and worst-case scenarios. **c** Per-capita excess mortality per age-group according to the transmission model (green) and observed data (red). The green line shows the model output for the median excess mortality according to the central model fit and the shaded area shows the 95% confidence intervals for the model estimate. The red line shows the excess mortality estimates based on the linear regression model with central line and shaded area representing the median and 95% forecast intervals for weekly excess mortality. **d** Cumulative attack rate of SARS-CoV-2 per province as of 2021-10-22.

each country[4,32,33]. International comparisons of COVID-19 death tolls have subsequently relied on excess mortality due to under-reporting of COVID-19 deaths in many parts of the world[2]. This study is the first to formally explore the suitability of excess mortality as a proxy for estimating under-reporting of COVID-19 infections and deaths, confirming the information contained within reliable excess mortality statistics through the broad agreement with representative seroprevalence estimates. Additionally, this analysis lends further support to evidence that IFR patterns estimated in high income countries may under-estimate the IFR in settings with stretched healthcare capacity[27], as experienced in many provinces in Iran at periods during the pandemic[34,35]. Even though in some provinces we find that the transmission model with a lower estimated IFR provides a better fit to seroprevalence estimates, such a model offers a poor fit across most provinces and often results in epidemiologically unrealistic outcomes such that the modelled transmission dynamics outruns the pool of susceptible individuals. We also note that while some factors such as misidentification of baseline all-cause mortality and deaths attributable to COVID-19, particularly in the younger age-groups with fewer COVID-19-related deaths, can contribute to varied attack rates in some provinces, other factors such as nonrepresentative serosurveys, varied time to seroreversion using different assays, and uncertainty in inferring the true contact matrix between different age-groups within the population can impact the inferred attack rates[36,37]. Absence of accurate vaccination data could be another potential source of bias that may influence our modelled attack rates in some provinces, particularly in provinces that experienced high transmission levels due to Delta during the last wave. For instance, if we overestimate the level of protection from vaccines, more individuals need to be infected in a given wave in order to produce an epidemic that correctly matches the observed mortality. Similarly, if more vaccines are distributed to younger individuals than we currently assume in the model, it could lead to an artificial increase in the IFR with those highest at risk not being protected.

Prior to the emergence of the variants of concern, the basic reproduction number of the SARS-CoV-2 virus was estimated to be around 2.8 which corresponds to a herd immunity threshold of about 64%[38]. This threshold does not represent the point at which the outbreak immediately stops, but rather the point where, on average, fewer secondary infections are produced. During unmitigated outbreaks with such reproduction numbers, the attack rate can reach as high as 90%[39,40]. Similarly, during unmitigated outbreaks with variants of concerns such as the Delta variant with increased reproduction numbers compared to the ancestral SARS-CoV-2 virus, the attack rate can reach even higher levels[41]. While the duration of antibody responses after coronavirus infection over longer timescales remains largely unknown, recent surveillance data and comparative studies have shown that re-infection with SARS-CoV-2 is possible after 3–6 months, and that older age-groups have a much lower protection against reinfection[42,43]. Previous studies on human coronaviruses and SARS-CoV have also shown that protective immunity typically starts declining after 5–8 months post infection[44–46]. Our findings of high attack rates in several provinces show that herd immunity through natural infection has not been achieved in the population even after nearly 20 months since the start of the Iranian epidemic. This is likely due to substantial reduction in protection against repeat infection over time either due to waning immunity, increased chance of re-infection with variants of concern, or a combination of both. We note that while systematic biases such as identifying the baseline level of all-cause mortality and overestimation of deaths attributable to COVID-19, particularly in the younger age-groups, can contribute to increased attack rates in some provinces, other

factors such as heterogeneities in immune protection, limited access to health care in municipalities with low socioeconomic status, and uneven adoption of public health intervention in different regions can also contribute to varied attack rates across the country.

## Methods

**Excess mortality**. We collect the weekly time-series data on all-cause mortality per province per age group from NOCR. We use data from the beginning of year 1394 to the end of summer 1398 in Solar Hijri calendar (SH) (from 2015-03-15 to 2019-09-22 in Common Era) to calculate background mortality. We find significant levels of excess mortality during several weeks (mostly concentrated in weeks 35 to 45) of autumn 2019 prior to the start of the COVID-19 pandemic (Supplementary Fig. 7). This has also been reported in a previous study which investigated some of the potential underlying causes of this significant excess mortality[15]. Therefore, in order to avoid biasing our expected mortality estimates from years prior to the pandemic, we exclude mortality data during autumn 2019 (from 2019-09-23 to 2019-12-21) from the calculation of background mortality as it may shift the baseline mortality upwards and, hence, result in the underestimation of excess mortality during the pandemic years in autumn 2020 and 2021. We calculate the expected mortality using a linear regression model that accounts for seasonality and trend, previously developed to track excess mortality across more than 100 countries and territories around the world[1]. As a robustness test, we also calculate expected mortality with the same model under the assumption of an over-dispersed Poisson to see if different methods of calculating excess mortality yields significantly different results. We find that the estimated excess mortality using both methods are virtually identical over time across all provinces (Supplementary Fig. 8). For example, we find that by 2021-10-22, the estimated total excess mortality at the national level based on the linear regression is 259,641 (95% CI: 230,493–288,790) persons while, based on the quasi-Poisson model, it is 258,472 (95% CI: 230,369–286,574) persons which shows that the two estimates are very similar to each other.

**Transmission model**. We use a previously published age-structured COVID-19 transmission model[47] and fitting framework[25] to fit the weekly estimated excess deaths in each province in Iran. In overview, the model is a population-based age-structured Susceptible-Exposed-Infected-Recovered model, which explicitly represents disease severity, passage through different healthcare levels and the roll out of vaccination. Model fitting was carried out within a Bayesian framework using a Metropolis-Hastings Markov Chain Monte Carlo (MCMC) based sampling scheme, which estimates the epidemic start date, $R_0$ and the time varying reproduction number, $R_t$, using a series of pseudo-random walk parameters, which alter transmission every 2-weeks, given by:

$$R_t = R_0.f(-\rho_1 - \rho_2 \cdots - \rho_n) \qquad (1)$$

where $f(x) = 2.exp(x)/(1 + exp(x))$, i.e., twice the inverse logit function. Each parameter is introduced two weeks after the previous parameter, serving to capture changes in transmission every two weeks. The last change in transmission, $\rho_n$, is maintained for the last 4 weeks prior to the current day to reflect our inability to estimate the effect size of this parameter due to the approximate 21-day delay between infection and death[22]. The MCMC sampling scheme includes adaptive tuning of the proposal implemented during sampling using the Johnstone-Chang optimisation algorithm[48]. All parameter inference results are based on 100,000 iterations, 10,000 of which are discarded as burn-in. MCMC chains are checked for convergence using standard Gelman-Rubin diagnostics[49].

Each model fit used province-specific demography, with the population size in 5-year age bands sourced from the Statistical Center of Iran[50]. The effective number of general hospital beds and intensive care beds were sourced from the Iran statistical yearbook published in 1396 in Solar Hijri calendar[51]. At the time of writing, MoHME has not published subnational daily vaccination data prior to 2021-10-26, nor the proportion of which vaccines are administered. As a result, we used multiple data sources to approximate the roll of vaccines subnationally. From 2021-10-26, MoHME started to release the total number of first and second vaccine doses subnationally, which we use in combination with national vaccination data[52] to infer the daily vaccination rate per province. A mix of vaccine types have been administered in Iran, with 80% of the population vaccinated with Sinopharm, 10% with AZ and the remaining 10% either Sputkink-V (1.5%) and CoViran Barekat (8.5%). Due to the mix of vaccine types with uncertain efficacy estimates, particularly related to Delta, we use vaccine efficacy estimates for first and second dose efficacy sourced from estimates for the Sinopharm vaccine and other inactivated whole virus vaccine platforms (Table 2). Efficacy estimates are weighted to produce an overall population efficacy by the proportion of vaccinated individuals who have received either a first dose or second dose, and by the assumed proportion of the Delta variant in the population, sourced from the sampling dates of Delta sequences in Iran from GISAID (gisaid.org), lagged by 14 days to reflect the delay from administering vaccine to protection being conferred.

Imposing a single IFR across the whole country may lead to biased attack rate estimates as it does not account for various factors such healthcare capacity or

**Table 2 Vaccine efficacy estimates, and parameter uncertainty related to hospitalisation and the Delta variant.**

| Description\Scenarios | Optimistic | Central | Worst | Notes & references |
|---|---|---|---|---|
| Vaccine efficacy against infection | Alpha: 50%; 67% Delta: 10%; 60% | Alpha: 50%; 67% Delta: 10%; 60% | Alpha: 50%; 67% Delta: 10%; 60% | 50%; 67% \| Dose 1; Dose 2 [55-57] |
| Vaccine efficacy severe disease | Alpha: 50%; 80% Delta: 20%; 70% | Alpha: 50%; 80% Delta: 20%; 70% | Alpha: 50%; 80% Delta: 20%; 70% | 50%; 80% \| Dose 1; Dose 2 [55-57] |
| Immune escape by the Delta variant | 10% | 27% | 50% | Median immune escape estimated at 27% with 50% Bayesian Credible Interval spanning 10%-50% [58] |
| Increased risk of hospitalisation from infections with the Delta variant | 1.08 | 1.45 | 1.95 | Increased probability of hospital—adjusted HR 1.45 (95% 1.08, 1.95) [59]—or emergency care attendance due to Delta variant within 14 days informed by UK Public Health England data. |
| Hospital Capacity | No shortage of beds at any point. | Bed numbers available equal to estimated bed capacity. | 50% of estimated total beds available due to occupancy with non-COVID-19 patients. | Total bed numbers sourced from Iran statistical yearbook published in 1396 in Solar Hijri calendar [51]. Baseline occupancy previously estimated at ~70% [60], which during COVID-19 we assume will be lower to reflect pandemic prioritisation as observed in other countries [61]. |

demographics of each province. To avoid this, we assume that the IFR in each province is dependent on the demographics of the province, such that provinces with older populations have, on average, higher IFRs. The model also tracks the number of people who require a general hospital bed or an ICU bed so that if there is a bed shortage during an epidemic peak in a given province then the mortality outcomes for individuals who do not receive a bed will increase. We explore three sets of model assumptions, which scan across the likely range of parameter estimates for key model parameters that impact the inferred attack rate. These parameters relate to the assumed number of general and ICU hospital beds available, the change in hospitalisation related to the Delta variant and the level of immune escape conferred by the Delta variant relative to previous infection by the Alpha variant and the original wild type strains. Here, the level immune escape only refers to immunity from previous infection, which acts by making a proportion of people who would have been protected against reinfection by the same variant susceptible to infection with Delta. With regards to immunity from vaccination, we assume that the impact of Delta is to reduce the level of protection conferred by the vaccines. These ranges are described in Table 2. Given uncertainty in the IFR and its impact of the inferred attack rate, we explore two alternative estimates of IFR by age[21,53] and conduct model comparison between the two references through comparison against province level estimated seroprevalence data[16], with seroprevalence assumed to be described by a Binomial distribution. Lastly, to explore if IFR is changing over time, we explore the ratio of model inferred hospitalisations and observed daily hospitalisations against time using a mixed-effects model, controlling for variation between provinces using a random intercept and slope with respect to time.

To assess the impact of healthcare capacity constraints on increasing the mortality from COVID-19, we also calculate the mean population-weighted IFR[15],

$$\alpha = \sum_{i=1}^{L} \alpha_i w_i,$$ and its corresponding standard error, $\sigma_\alpha = \sqrt{\sum_{i=1}^{L} w_i^2 \sigma_i^2}$, for each

province using the estimates of IFR by age from Brazeau et al.[53], where $L$ is the total number of age-groups, $\alpha_i$ is the age-stratified IFR for age-group $i$, $\sigma_i$ is its standard error, and $w_i$ is the proportion of individuals within age-group $i$ in the population. Unlike IFR estimates based on our transmission model which takes into account the impact of various factors such as province-level demographics, impact of healthcare capacity constraints on increased mortality rates, and increased chance of hospitalisation with Delta the population-weighted IFR only accounts for the province-level demographics which enables us to assess the influence of other factors on increasing IFR.

**Reporting summary**. Further information on research design is available in the Nature Research Reporting Summary linked to this article.

## Data availability

Raw weekly mortality data files in Iran are retrieved from https://www.sabteahval.ir/avej/Page.aspx?mId=49826&ID=3273&Page=Magazines/SquareshowMagazine Number of active beds in hospital wards by province in year 1396 Solar Hijri (2018) is available at https://iranopendata.org/en/dataset/number-of-active-beds-in-hospital-wards-by-province-in-year-1396/resource/73e29d0b-7cc9-47c0-8b4e-b3e61cc0b800 Number of administered vaccine doses per province is retrieved from the official Telegram channel of the Statistics and Information Technology Center of the Ministry of Health and Medical Education of Iran https://t.me/it_behdasht/134 Daily hospital admission data is collected by the Ministry of Health and Medical Education of Iran. The data as of 22 November 2021 is retrieved from the mask.ir application official telegram channel https://t.me/mask_application/430 Province-specific demography, with the population size in 5-year age bands, is retrieved from the Statistical Center of Iran website https://www.amar.org.ir/english/Iran-Statistical-Yearbook/Iran-Statistical-Yearbook-2019-2020.

## Code availability

All software code and analysis scripts are available in an R research compendium at https://github.com/OJWatson/iran-ascertainment[54].

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

## Acknowledgements
M.G. is funded by the Biotechnology and Biological Science Research Council (BBSRC), grant number BB/M011224/1. O.J.W. is supported by a Schmidt Science Fellowship in partnership with the Rhodes Trust. A.Kat is supported by the European Research Council, grant number 101001623-PALVIREVOL.

## Author contributions
Conceptualisation: M.G., A.Kat, O.J.W. and L.F.; Methodology: M.G., O.J.W. and A.Kar; Investigation: M.G. and O.J.W.; Visualisation: M.G., O.J.W. and A.Kar; Supervision: A.Kat; Writing—original draft: M.G.; Writing—review & editing: All authors.

## Competing interests
The authors declare no competing interests.
