## [Peer Review File · Nature Communications]

A framework for reconstructing SARS-CoV-2 transmission dynamics using excess mortality dataREVIEWER COMMENTS

Reviewer #1 (Remarks to the Author):

Review of NCOMMS-22-00215-T

Reliable COVID mortality data are in short supply worldwide. There have been numerous studies using excess mortality from civil registration systems to infer COVID-19 mortality and infection fatality ratios. In this clearly written paper based on excess mortality data from Iran, the authors propose a method to match representative seroprevalence estimates, with those derived from excess mortality used as a proxy for estimating under-reporting of COVID-19 infections and deaths. The alignment between excess mortality and seroprevalence estimates is notable. However, the lack of information on vaccination coverage and rollout is a significant weakness and the implications of this should be discussed in greater detail.

Overall, it's a nice piece of work and contribution to the literature. A few points to consider.

1. While IFRs in Iran may certainly be higher than in high-income countries, I would be skeptical that they are far higher despite the point about healthcare infrastructure being weaker. Did the authors consider heterogeneity in health care capacity across Iran and account for this in some way? There are parts of the paper that seem to suggest this but it is unclear from the methods section. Imposing a single IFR across the entire country may be unrealistic and indexing this to some measure of healthcare capacity (perhaps hospital beds per capita as a proxy), may improve model fit.
2. Line 233: I think you mean "rollout" of vaccines.
3. Line 236: Sputnik is misspelt.
4. Table 1: Golestan has an exposure rate of 105%. Seems unlikely – is this a calculation bug?
5. On assumptions – does the immune escape parameter refer to immunity through prior exposure or vaccination? Also, since multiple vaccines have been used, what is the source of the single numbers for vaccine efficacy.
6. On figure 2, it would be helpful to have the three panels lined up and stacked so that one can clearly see the excess deaths, concentrated in the periods of the peaks.

Reviewer #2 (Remarks to the Author):

I read the manuscript with enthusiasm, because it deals with an important issue, which concerns transmission models of COVID-19 in Iran and its regions. A country that since 2020 has been strongly affected by this new illness. Before its publication, I advise the authors to consider the following comments.

General comments

- Lines 121-122. The population-weighted IFRs, seems that the authors estimate this measure indirectly using information from high-income countries. Needs more clarification about this indicator. In the section of methods there is no information about it.
- Lines 207-210. We exclude mortality data during autumn 2019 (from 2019-09-23 to 2019-12-21) from the calculation of background mortality as we have previously reported elevated mortality rates across several provinces, unrelated to the COVID-19 pandemic, which can bias our estimates for the expected mortality.
I did not understand this procedure. The authors mentioned that according to other studies, the pandemic in Iran may started in late December 2019 (line 79). Hence, I agreed to remove this month from the estimates of excess mortality, but why previous months? I read the paper mentioned that identified the spikes of mortality from September to December 2019. Could be this not related to other data issues (later registration for instance)?

- Lines 210-212. The estimation of excess mortality uses a linear regression model. How this model performs in comparison with other ways to estimate excess mortality? For example, Poisson or quasi-Poisson, see:
 - Lucas Sempé, Peter Lloyd-Sherlock, Ramón Martínez, Shah Ebrahim, Martin McKee, Enrique Acosta, Estimation of all-cause excess mortality by age-specific mortality patterns for countries with incomplete vital statistics: a population-based study of the case of Peru during the first wave of the COVID-19 pandemic, *The Lancet Regional Health - Americas*, Volume 2, 2021. <https://doi.org/10.1016/j.lana.2021.100039>.A critical discussion about these estimates is desirable.

- Lines 215-216. The authors applied a SEIR model, but is it possible to compare this estimate with a simple SIR model? I recommend a critical discussion about this model, see
 - Roda, W. C., Varughese, M. B., Han, D., & Li, M. Y. (2020). Why is it difficult to accurately predict the COVID-19 epidemic?. *Infectious Disease Modelling*, 5, 271–281. <https://doi.org/10.1016/j.idm.2020.03.001>
 - Sánchez-Romero, Miguel, et al. "An indirect method to monitor the fraction of people ever infected with COVID-19: An application to the United States." *PloS one* 16.1 (2021): e0245845.
 - Sánchez-Romero, Miguel, et al. How many lives can be saved? A global view on the impact of testing, herd immunity and demographics on COVID-19 fatality rates. No. 05/2020. *ECON WPS*, 2020.
 - Berger, D., Herkenhoff, K., Huang, C. and Mongey, S., 2020. Testing and reopening in an SEIR model. *Review of Economic Dynamics*.
 - Wang, Jin. "Mathematical models for COVID-19: applications, limitations, and potentials." *Journal of public health and emergency* 4 (2020).
 - Gibson, Gavin J., George Streftaris, and David Thong. "Comparison and assessment of epidemic models." *Statistical Science* 33.1 (2018): 19-33.

- The MCMC needs more clarification.

Minor comments

- For the ratio estimate, why the use of the group 40-49 years old?
- Figure 2(C). The text mention areas in black, but it is not seen in the correspondent figure.
- Figure S1. The black lines representing the observed values are absent for the provinces.

Reviewer #1 (Remarks to the Author):

Review of NCOMMS-22-00215-T

Reliable COVID mortality data are in short supply worldwide. There have been numerous studies using excess mortality from civil registration systems to infer COVID-19 mortality and infection fatality ratios. In this clearly written paper based on excess mortality data from Iran, the authors propose a method to match representative seroprevalence estimates, with those derived from excess mortality used as a proxy for estimating under-reporting of COVID-19 infections and deaths. The alignment between excess mortality and seroprevalence estimates is notable. However, the lack of information on vaccination coverage and rollout is a significant weakness and the implications of this should be discussed in greater detail.

Overall, it's a nice piece of work and contribution to the literature. A few points to consider.

We thank the reviewer for their constructive feedback and positive assessment of our work. We agree with the reviewer's assessment that information on vaccination coverage and rollout is lacking. We would like to highlight that this information is not regularly updated by the ministry of health of Iran (e.g., see our opinion piece on this issue: <https://blogs.bmj.com/bmj/2021/08/03/irans-covid-19-vaccination-programme-using-transparency-to-build-public-trust-in-immunisation/>).

We also note that an increasing number of administered vaccines in Iran are from the CoVIran Barekat for which there are no clinical studies on efficacy against infection, hospitalisation, and deaths. However, we note that during the period over which we conducted our study, the majority (>80%) of the population have received only one vaccine type. We have now discussed this limitation and its implications for our analysis in more detail in the Discussion. We have also added a supplementary figure, showing the timeline of vaccination coverage per age group in Iran.

1. While IFRs in Iran may certainly be higher than in high-income countries, I would be skeptical that they are far higher despite the point about healthcare infrastructure being weaker. Did the authors consider heterogeneity in health care capacity across Iran and account for this in some way? There are parts of the paper that seem to suggest this but it is unclear from the methods section. Imposing a single IFR across the entire country may be unrealistic and indexing this to some measure of healthcare capacity (perhaps hospital beds per capita as a proxy), may improve model fit.

We thank the reviewer for highlighting this key point and apologise for not having made this clearer. We have indeed incorporated this in our initial analysis but now also make this clearer in the abstract and discussion.

Our model assumes that the IFR in each province is dependent on the demographics of the province, with provinces with older populations having higher IFRs on average, with the

specific relationship between IFR and age either taken from Brazeau et al or O’Driscoll et al. However, the model also tracks the number of people who require a general hospital bed or an ICU bed and if there is a bed shortage then the mortality outcomes for individuals who do not receive a bed will be increased. We use best estimates for the number of ICU beds and general hospital beds for each province, which are sourced from the Iran statistical yearbook published in 1396 in Solar Hijri calendar. Consequently, our province-level IFR estimates take into account the impact of hospital bed capacity on changing IFR. That is why our estimated IFR is largely different from the population-weighted IFR which only takes into account the demographics of a given province (see columns 4 and 5 in Table 1). We have now highlighted this in the Results section and to make the methodological approach taken clearer, we have added additional text to the Methods to make this process clearer.

2. Line 233: I think you mean “rollout” of vaccines. -- corrected

3. Line 236: Sputnik is misspelt. -- corrected

4. Table 1: Golestan has an exposure rate of 105%. Seems unlikely – is this a calculation bug?

Thank you for highlighting Golestan. We have checked over the calculations and this is not a bug. Golestan had one of the highest per-capita excess deaths in the first wave and has since experienced four additional waves throughout 2020-21. In particular, the latest wave which was dominated by Delta led to the highest level of excess mortality for this province since the beginning of the pandemic. As a result, our estimates indicate that this province had one of the highest levels of re-infection and attack rate compared to other provinces, which allowed the exposure rate to be above 100%.

However, it is possible that this attack rate is overestimated. We discuss a number of possible limitations to the model approach in the Discussion, which could contribute to an overestimation. One possible explanation, which we did not explore in the Discussion, is that we may be overestimating the protective effect caused by vaccines. If vaccine protection is high, more people need to be infected during the last wave in order to produce an epidemic that correctly matches the observed mortality. Similarly, if more vaccines were distributed to younger individuals than we assume, this would also lead to an increase in the IFR with those highest at risk not being protected. However, as you noted in one of your earlier comments, accurate data on vaccination coverage is not available. In response, we have added extra text to the Discussion to highlight that the absence of accurate vaccination data may also be impacting our inferred attack rates.

5. On assumptions – does the immune escape parameter refer to immunity through prior exposure or vaccination? Also, since multiple vaccines have been used, what is the source of the single numbers for vaccine efficacy.

Thank you for raising these points. Immune escape here refers to immunity from previous infection, which acts by making a proportion of people who would have been protected against reinfection by the same variant now susceptible to infection with Delta. We have added an

additional sentence to the Methods to make this clearer. With regards to immunity from vaccination, we assume that the impact of Delta on the protection conferred by vaccines is modelled by reducing the level of protection that the vaccine confers. This is currently described in the manuscript but have added additional clarification to the Methods.

Lastly, the single numbers for vaccine efficacy have been used for two reasons. Firstly, the model used has one compartment that tracks individuals who were vaccinated more than 14 days ago (i.e. the immunity they have gained from vaccination is likely to have formed). However, there is a mix of vaccines being used as well as a mix of individuals who will have received one or two doses. We account for the mix of doses, by weighting the vaccine efficacy by the assumed number of people who have received two vs one doses. However, this is just an assumption based on national data as we do not have any data at the province level. This absence of data, combined with not knowing the true mix of which vaccines have been used (and what the vaccine efficacy is of the Iranian Barekat vaccine) means any number we use will be an assumption. Given that 80% have received Sinopharm, we have based our vaccine efficacy on this vaccine. This is a limitation and may likely be overestimating vaccine efficacy if Barekat has a much lower vaccine efficacy (this may be one reason for attack rates as estimates being an overestimate during later waves). We still believe the current approach is the best assumption we can make with the data available, however, we have added more text to the Discussion to highlight these limitations and explore the impact they may have on our overall findings.

6. On figure 2, it would be helpful to have the three panels lined up and stacked so that one can clearly see the excess deaths, concentrated in the periods of the peaks.

On figure 2, we are not showing any analysis of excess deaths. We think perhaps the reviewer is referring to the panels in figure 1 which show the nationwide and province-level excess mortality. We have now aligned and stacked them as suggests.

Reviewer #2 (Remarks to the Author):

I read the manuscript with enthusiasm, because it deals with an important issue, which concerns transmission models of COVID-19 in Iran and its regions. A country that since 2020 has been strongly affected by this new illness. Before its publication, I advise the authors to consider the following comments.

We thank the reviewer for this positive assessment of our work. We also concur that while Iran was one of the countries to be significantly impacted by an early wave of COVID-19 in 2020, there has been very limited number of studies trying to examine the transmission dynamics across the country. That is why our newly developed method is particularly relevant for countries like Iran where there is limited epidemiological and clinical data on COVID-19 transmission.

General comments

- Lines 121-122. The population-weighted IFRs, seems that the authors estimate this measure indirectly using information from high-income countries. Needs more clarification about this indicator. In the section of methods there is no information about it.

We agree and thank the reviewer for bringing this to our attention. We added this IFR measure to Table 1 (see column #5) based on a previous study by Ghafari et al ([10.1016/j.ijid.2021.04.015](https://doi.org/10.1016/j.ijid.2021.04.015)) on the population-weighted IFR estimates in Iran. In that study, the population-weighted IFR was estimated using the age to IFR relation from the O'Driscoll et al (analysis based mostly on high-income countries) which has much lower IFR estimates compared to the Brazeau et al study. Given that our estimated IFR in the current study (Table 1, column #4) is based on the age to IFR relation from Brazeau et al (which we found was more accurate based on our model fits to seroprevalence data in each province), we have now re-calculated the population-weighted IFRs to allow both estimates to be compared relative to the same age to IFR reference point (i.e., Brazeau et al).

The significance of showing the population-weighted IFRs for each province is that it allows us to compare the estimated IFR based on our transmission model (which accounts for various factors impacting IFR including healthcare capacity constraints, population demography, and increased chance of hospitalisation with Delta) with the population-weighted IFR estimate (which only accounts for demography). Our results suggest the IFR estimates based on the former are much higher than the latter, indicating that factors such as hospital bed capacity increasing mortality rates have influenced our net IFR estimates. We have now provided more information on population-weighted IFR in the Methods and have highlighted this comparison in more detail in the Results.

- Lines 207-210. We exclude mortality data during autumn 2019 (from 2019-09-23 to 2019-12-21) from the calculation of background mortality as we have previously reported elevated mortality rates across several provinces, unrelated to the COVID-19 pandemic, which can bias our estimates for the expected mortality.

I did not understand this procedure. The authors mentioned that according to other studies, the pandemic in Iran may started in late December 2019 (line 79). Hence, I agreed to remove this month from the estimates of excess mortality, but why previous months? I read the paper mentioned that identified the spikes of mortality from September to December 2019. Could be this not related to other data issues (later registration for instance)?

A previous study by Ghafari et al ([10.1016/j.ijid.2021.04.015](https://doi.org/10.1016/j.ijid.2021.04.015)) showed that there were significant levels of excess mortality across Iran during autumn 2019, unrelated to the pandemic. To avoid biasing our expected mortality estimates from years prior to pandemic, we removed this period from our calculation as it shifts the baseline upwards and, hence, may result in underestimation of excess deaths related to COVID-19 in autumn 2020 and 2021.

We apologise if the procedure was not clear and have now clarified this in the methods section. We have also included a supplementary figure showing elevated levels of mortality during

autumn 2019 in almost all provinces across the country. Understanding the underlying cause(s) of the elevated deaths in autumn 2019 has been extensively discussed in the previous study by Ghafari et al and we believe it falls outside the scope of the current study.

• Lines 210-212. The estimation of excess mortality uses a linear regression model. How this model performs in comparison with other ways to estimate excess mortality?

For example, Poisson or quasi-Poisson, see:

- Lucas Sempé, Peter Lloyd-Sherlock, Ramón Martínez, Shah Ebrahim, Martin McKee, Enrique Acosta, Estimation of all-cause excess mortality by age-specific mortality patterns for countries with incomplete vital statistics: a population-based study of the case of Peru during the first wave of the COVID-19 pandemic, *The Lancet Regional Health - Americas*, Volume 2, 2021. <https://doi.org/10.1016/j.lana.2021.100039>.

A critical discussion about these estimates is desirable.

We thank the reviewer for bringing these references to our attention. We have now conducted the analysis of excess mortality using a quasi-Poisson method for all 31 provinces and showed that estimated excess mortality using our linear regression model and a quasi-Poisson model are virtually identical. For example, we find that by 2021-10-22, the estimated cumulative excess mortality at the national level based on the linear regression is 259,641 (95% CI: 230,493 - 288,790) persons while, based on the quasi-Poisson model, it is 258,472 (95% CI: 230,369 - 286,574) persons which indicates that the two estimates are very similar to each other. Therefore, we believe the method of choice for calculating excess mortality is not going to significantly impact our findings.

• Lines 215-216. The authors applied a SEIR model, but is it possible to compare this estimate with a simple SIR model?

I recommend a critical discussion about this model, see

- Roda, W. C., Varughese, M. B., Han, D., & Li, M. Y. (2020). Why is it difficult to accurately predict the COVID-19 epidemic?. *Infectious Disease Modelling*, 5, 271–281. <https://doi.org/10.1016/j.idm.2020.03.001>

- Sánchez-Romero, Miguel, et al. "An indirect method to monitor the fraction of people ever infected with COVID-19: An application to the United States." *PloS one* 16.1 (2021): e0245845.

- Sánchez-Romero, Miguel, et al. How many lives can be saved? A global view on the impact of testing, herd immunity and demographics on COVID-19 fatality rates. No. 05/2020. *ECON WPS*, 2020.

- Berger, D., Herkenhoff, K., Huang, C. and Mongey, S., 2020. Testing and reopening in an SEIR model. *Review of Economic Dynamics*.

- Wang, Jin. "Mathematical models for COVID-19: applications, limitations, and potentials." *Journal of public health and emergency* 4 (2020).

- Gibson, Gavin J., George Streftaris, and David Thong. "Comparison and assessment of epidemic models." *Statistical Science* 33.1 (2018): 19-33.

Thank you for the comment and the references listed. The exposed compartment in our model will not affect the model results, which are determined by the model fits that compare the model outputted deaths to excess deaths in each province. The simple SIR models that are detailed in the references would not be suitable for the analysis we have undertaken, which relies on both tracking the number of people in general hospital beds and ICU beds, as well as those who may not receive a bed if there is a shortage and how this impacts the fatality rate over time. All of these extra compartments are contained within our I compartment, which itself is age disaggregated to correctly account for age dependent infection fatality rates and age dependent mixing among the population. In addition, we also have compartments for tracking each of these different states with respect to vaccination status.

With regards to the references listed, the comparison between a simple SIR (3 compartments) and SEIR (4 compartments) when fitting to case data, the SIR model will always outperform the SEIR model based on AIC characteristics, as the exposed compartment simply functions as a latent term, delaying the flow from S to I. However, the reason you would include an exposed compartment is not for formal model comparisons to reproduce case data but to consider other questions that require it, for example exploring case isolation. The reason we have an exposed compartment is simply that it was encoded in the model for previous analyses and studies by the group to look at questions related to testing and case isolation. We could remove it but it would not affect the model likelihood based on the fit to excess deaths and would not add any further insight to the key questions of this study, which relate to the use of excess death data to understand the Iranian epidemic in each province.

- The MCMC needs more clarification.

Thank you for raising the need for more clarity. We have expanded the description of the MCMC related to the tuning, chain lengths, burn-ins and assessments of chain convergence to allow the reader to understand in overview the MCMC. Full MCMC details are detailed in the Methodology of the cited manuscript (Watson et al. 2021. Nat Comms), which uses the same approach as in this manuscript.

Minor comments

- For the ratio estimate, why the use of the group 40-49 years old?

The ratio estimate can be used based on other age-groups as well. The reason for choosing 40-49 is partly because unlike younger age-groups it is subject to less stochasticity over time (due to higher deaths relative to younger age-groups) and also serves as a baseline for measuring the impact of vaccination on lowering excess deaths in older age-groups who have been vaccinated first. We have now clarified this in the text and added a supplementary figure showing the timeline for vaccination in different age-groups which also shows that 40-49 age group started getting vaccinated at a much later date compared to older age-groups.

- Figure 2(C). The text mention areas in black, but it is not seen in the correspondent figure.

The shaded area in black is referring to panels A and B (not C). We have now corrected the text.

- Figure S1. The black lines representing the observed values are absent for the provinces.

Thank you for this comment. In Figure 1A, we have a black line in the plot, which represents the nationally reported COVID-19 deaths, which we compare to the excess mortality in blue. However, in Figure S1, there is no black line to plot as COVID-19 deaths are not reported at the province level. We have made this distinction clearer, by noting in the legend for Figure S1 that subnational data is not released (this was one of the reasons behind this manuscript to better understand subnational patterns given that the government has not been releasing this data).